# Endothelial Dysfunction Induced by Extracellular Neutrophil Traps Plays Important Role in the Occurrence and Treatment of Extracellular Neutrophil Traps-Related Disease

**DOI:** 10.3390/ijms23105626

**Published:** 2022-05-17

**Authors:** Shuyang Yu, Jingyu Liu, Nianlong Yan

**Affiliations:** Department of Biochemistry and Molecular Biology, School of Basic Medical Science, Nanchang University, Nanchang 330006, China; ysydoctor@163.com (S.Y.); liujingyu000726@126.com (J.L.)

**Keywords:** neutrophil, extracellular neutrophil traps (NETs), endothelial dysfunction (ED), inflammation, damage-associated molecular patterns (DAMPs), treatment

## Abstract

Many articles have demonstrated that extracellular neutrophil traps (NETs) are often described as part of the antibacterial function. However, since the components of NETs are non-specific, excessive NETs usually cause inflammation and tissue damage. Endothelial dysfunction (ED) caused by NETs is the major focus of tissue damage, which is highly related to many inflammatory diseases. Therefore, this review summarizes the latest advances in the primary and secondary mechanisms between NETs and ED regarding inflammation as a mediator. Moreover, the detailed molecular mechanisms with emphasis on the disadvantages from NETs are elaborated: NETs can use its own enzymes, release particles as damage-associated molecular patterns (DAMPs) and activate the complement system to interact with endothelial cells (ECs), drive ECs damage and eventually aggravate inflammation. In view of the role of NETs-induced ED in different diseases, we also discussed possible molecular mechanisms and the treatments of NETs-related diseases.

## 1. Introduction

The vascular endothelial barrier plays a vital role in the defense of foreign infections and injury, serving as a boundary between blood and tissue [1]. However, in the circulation, neutrophils are confirmed to have a significant impact on endothelial function by directly damaging endothelial cells (ECs) and their surrounding structures [2,3]. Furthermore, neutrophils also can release a particular structure termed extracellular neutrophil traps (NETs), which can induce the local inflammation of ECs, resulting in the endothelial dysfunction (ED). ED is involved in many diseases, including cardiovascular disease (CVD), Autoimmune Diseases, Systemic Lupus Erythematosus (SLE), Sepsis, diabetic retinopathy (DR), etc. Moreover, increased studies have discovered some definite mechanisms of NETs, and the common points are inflammation- and drugs research-related. Hence, the review focuses on the relationship between neutrophil-derived NETs and ED, especially with inflammation as a standpoint to explore the mechanism of ED, which is related with the occurrence and treatment for NETs-related diseases by targeting the pathophysiological processes of NETs.

## 2. The Relationship between NETs and ED

### 2.1. What Is Neutrophil Extracellular Traps

Neutrophils are the richest cell species of white blood cells in blood circulation, playing a vital part in human homeostasis [4]. They are termed on account of the colorless cytoplasm, with many scattered reddish particles in Wright-stained blood smears, and they have particular lobulated structures of nucleus, which is why they are also called polymorphonuclear cells (PMNs) [4]. Moreover, neutrophils are differentiated from hematopoietic stem cells, and further depart from bone marrow to defend against infections. The most familiar function of them is antimicrobial capacity-defensing against the microbial pathogens on the first line with a powerful antimicrobial arsenal in their granules [5], which exert an important effect in the nonspecific cellular immune system of blood. Interestingly, recent studies have suggested that neutrophils may also have a memory-like immune mechanism against infection, termed neutrophil adaptive (memory-like) reactions [6,7,8,9]. For instance, Bacillus Calmette-Guerrin (BCG) vaccine and β-glucan can induce the adaptive response of naive neutrophils [7,8], which leads to reprogramming of neutrophil transcriptome and epigenetics modifications in trimethylation at histone 3 lysine 4, and finally the release of pro-inflammatory mediators. As a result, when neutrophils are similarly stimulated for the second time, they can be heavily recruited and activated to mediate a more intense immune response. Meanwhile, an interesting study found that the process may be concentration-dependent [6,9]. Low doses of microbiota-derived components (LPS or Small Extracellular Vesicles) have been shown to induce adaptive response of neutrophils, while high doses of microbiota-derived components can inhibit it [9]. Mechanistically, alterations in TLR2/MyD88, as well as TLR4/MyD88 signaling, were correlated with the induction of adaptive cues in neutrophils in vitro. However, during pathogen invasion, the neutrophils can be activated and release NETs, which is strictly distributed by three programs: phagocytosis, degranulation and the release of NETs [5]. NETs are extracellular web structures composed by the bracket of the decondensed chromatins with the aggregation of cytoplasmic and granule proteins [10], and the DNA of NETs is mostly from the nucleus with slight mitochondria DNA [11]. Although NETs are important to some specific infections, neutrophils can also release NETs under pathological conditions or in vitro stimulation [11]. Therefore, NETs may defend against infections, and also hurt the host tissue by damaging the host homeostasis.

### 2.2. The Formation of NETs

There are two types of NETs (Figure 1), and the release of NETs is mediated by the death of neutrophils called NETosis [12]. During NETosis (Figure 1), the dynamics of actin is restricted with neutrophils depolarization, then the uncondensed nuclear chromatins enter into the intact cytoplasm after the dissolution of the nuclear envelope [5,12]. Afterwards, the chromatins are mixed with cytoplasm and granules, and further expand towards the extracellular space when the cell membrane become permeabilized [12]. The other mechanism called non-lytic NETosis (Figure 1) performs the rapid liberation of NETs with the secretion of chromatins and granule components avoiding cell death [13,14]. The non-lytic mechanism is triggered by the first neutrophils arrived at the infectious sites [5]. Previous studies have shown that the phorbol 12 myristate 13 acetate (PMA)-induced ROS can initiate the NETs formation, and two enzymes within the ROS pathway play a crucial role during NETosis [15]. Following the formation (Figure 1), nicotinamide adenine dinucleotide phosphate (NADPH) oxidase catalyzed ROS stimulate myeloperoxidase (MPO), which further activates neutrophil elastase (NE) to translocate from azurophilic granules to the nucleus, resulting in proteolyzing the histone and destructing the chromatin assemble [16]. Subsequently, MPO binds to chromatin and decondenses them together with NE without enzymes catalysis [16]. During quiescent state, a portion of MPO termed azurosome combinea with NE, which ia selectively released to the cytoplasm through H_2_O_2_ in an MPO-dependent mode, and then the cytoplasmic NE bind and degrade F actins to enter into the nucleus [17]. Therefore, the oxidative activation is necessary for NETosis. Nevertheless, NADPH oxidase is unnecessary for NETs’ formation, since some stimuli can trigger NETosis through mitochondrial ROS, which is independent of NADPH oxidase [11]. Moreover, ROS not only decondense chromatins, it also reacts with hypochlorous acid from MPO to produce the chlorinated polyamines, which further crosslink the proteins of NETs, in order to facilitate NETs’ stability and improve the microbial trap capacity. The protein-arginine deiminase type 4 (PAD4) mediates another mechanism of chromatin decondensation through histone deamination or citrullination [18,19], as a nuclear enzyme citrullinates arginine residue in a reductive environment [20]. Moreover, H_2_O_2_ can stimulate PAD4 within the calcium condition, which is induced by PKC, a kinase of ROS pathway [21,22,23]. Therefore, it is supposed that PAD4 is sited in the downstream of ROS and calcium transduction during NETosis [5], and the citrullination depends on the isoform of Protein Kinase C (PKC) activated by various stimuli [23,24]. Moreover, different damages will generate different patterns to activate NETosis, but the most vital factor is inflammation [5]. In summary, the majority of NETosis is implicated in the inflammatory response.

### 2.3. Endothelial Dysfunction

Neutrophils-produced NETs can protect the host from pathogens including most gram-negative bacteria, gram-positive bacteria, viruses, parasites and so on [25]. First of all, it can prevent the spread of infections through the extracellular DNA network from NETs [10]. Second, it will kill pathogens through its own ingredients; for instance, NE plays a role through slicing the virulence factors [10]. However, in addition to host protection function, when it exists for a long time or accumulates in large quantities, it will initiate a variety of pathological processes related to endothelial damage, and eventually cause permanent damage to ECs [26]. Studies have shown that the histones MPO, NE and Cathepsin G are the primary components of NETs-engaged tissue destruction [27]. Mainly, NETs can affect the physiological function of ECs through inflammation. The ECs are a single layer of cells found in the lumen of the blood vessels, which functions as a physical barrier between circulating blood and the underlying materials. It is also a kind of cell system that plays a vital physiological role in maintaining vascular homeostasis. The ECs mainly regulate vascular tone, balance fibrinolysis and thrombogenesis, mediate the onset of inflammation and immune responses and promote the formation of new blood vessels [28,29,30]. Endothelial activation is a pro-inflammatory and pro-coagulant phenotype triggered by immune cell-secreted cytokines, which are exposed to inflammatory conditions. It is distinguished by the expression of cell surface adhesion molecules, chemokines and cytokines required for the recruitment and attachment of inflammatory cells on the vascular wall such as leukocytes [31,32,33]. Thus, the impairment of vascular endothelium function and other endothelial physiological functions above refers to ED. It inclines the vessels to a pro-thrombotic and pro-atherosclerotic state with features such as vasoconstriction, leukocyte adhesion, platelet activation, mitogenesis, pro-oxidation, coagulopathy, vascular inflammation, atherosclerosis (AS) and thrombosis [30,34]. Therefore, ED is highly relevant to a wide range of diseases including AS, diabetes, coronary heart disease, hypertension and hypercholesterolemia. Many abnormal vascular physiological states can induce ED to develop these diseases. In addition to the classic risk factors of oxidative stress and inflammatory response, other incentives have been identified in recent years, such as mental stress, ageing and exposure to specific drugs, especially the NETs [35]. 

### 2.4. The Linkages of NETs and ED

NETs not only contribute to the antimicrobial functions; they also are confirmed to connect with many pathologies of diseases, which mainly depend on the constitutive activation, dysregulation of suppressive mechanisms and excessive NETs’ yield [5]. Inflammation acts as a kind of defense mechanism in the case of injury-like infection, which is highly correlated with NETs. What is more, the inflammatory response can damage the endothelium through promoting the expression of ROS and NF-κB in the ECs, and further mediate the endovascular inflammation and platelets aggregation [36]. Furthermore, the systemic inflammatory response also can recruit neutrophils and generate NETs, which then promote the local inflammation of vascular endothelium. Therefore, it is accepted that NETs play a vital role during inflammatory response and even induce ED via amplifying local inflammation. 

#### 2.4.1. The Components of NETs Cause ED by Damage-Associated Molecular Patterns

The “danger” theory was proposed in 1994, which described the damaged cells initiating an immune response by releasing substances [37]. Walter Land later defined it as Damage-associated molecular patterns (DAMPs) in 2003 [38]. DAMPs play a significant role in the NETs-induced ED, which is associated with extracellular histone, extracellular cold-inducible RNA-binding protein (eCIRP) and high-mobility group box 1 (HMGB1), etc (Figure 2). Extracellular histones, as the most typical DAMPs, are released following NETosis and activate NF-κB and AP-1 pathways via Toll-like Receptor (TLR) on the EC membrane [39,40]. Studies published by Aldabbous and his colleagues have shown that the low concentrations of NETs can promote the attachment of histones to TLR4 and stimulate the release of inflammatory cytokines from ECs by activating the NF-κB pathway, ultimately resulting in ED [40]. With the increased NETs, histones from NETs can also bind to TLR2&4 receptors simultaneously to initiate the intracellular NF-κB pathway through the nuclear translocation of p65 and c-Rel, or directly activate AP-1 transcription, and further elevate the tissue factor (TF) expression in ECs. It facilitates the platelet aggregation and finally gives rise to ED [39]. Moreover, extracellular histones bind to the cell membrane and form pores to allow calcium ions to flow in, resulting in calcium overload, which directly damages the exposed cells [41,42]. For other molecules of DAMPs, only a few studies have reported that they can damage the ECs. eCIRP was originally thought to be a protein that inhibits mitosis and promotes cell differentiation in hypothermia [43]. Recent reports suggested that the excessive release of eCIRP from NETs upregulates the NLRP3 inflammasome through the TLR4/TLR9/NF-κB signaling pathway [44]. At the same time, they generate ROS to injure ECs [44]. Moreover, NLRP3 inflammasome can also be activated by the influx of calcium ions and ROS to promote oxidative stress and cause ED. Therefore, there may be a potential positive feedback facilitating inflammation and ED [45]. Moreover, other studies suggested that eCIRP can induce ER stress by stimulating NLRP3 inflammasome, which eventually leads to the pyroptosis of pulmonary ECs [46,47]. Moreover, it has been reported that the pyroptosis of ECs is also one of the pathophysiological processes of sepsis [48]. In addition, after NLRP3 was activated, ECs could secrete IL-1β, IL-18 and HMGB1 [45]. These particles can bind to EC surface receptors to activate NF- κB signal pathway and further promote the production of pro-inflammatory mediators (TGF-α, IL-6, IL-1β, IL-8) [45,49]. What is more, IL-8 can also form positive feedback by recruiting more neutrophils, thus intensifying the NETosis and inflammatory response on ECs [3,50,51]. Beyond that, HMGB1 has also been proved to be an inducement of aseptic inflammation [52]. Some researchers have suggested HMGB1 can activate macrophages and ECs produce EC adhesion molecules, chemical factors and cytokines [53,54]. However, it is still controversial whether HMGB1 has a harmful impact on ECs as one of DAMPs. More experiments are needed to prove the specific role of HMGB1 in NETs-induced ED. Moreover, the internalization of NET-bound RNA by ECs is induced type I IFN-stimulated genes and leads to ED, which depends on the endosomal TLR7 and the actin cytoskeleton [55]. From the previous discussion, it can be summarized that many molecules in NETs participate in ED as DAMPs, including histone, eCIRP and so on. More specific studies are required to prove the specific role of HMGB1 in NETs-induced ED. 

#### 2.4.2. Enzymes of NETs Can Cause the ED

Studies have shown NETs impact ECs through specific enzymes (Figure 3). Matrix metalloproteinase (MMP) are the main components of NETs, which are tightly related with ED. For example, NETs can increase the release and expression of MMP-9, which further elevates the MMP-2 level derived from ECs [56]. Carmelo has proved that the co-culture of anti-MMP-9, NETs and HUVECs would alleviate endothelial injury with the significantly reduced expression of MMP-9 and endothelial MMP-2 [56]. Therefore, MMP-9-activated MMP-2 leads to ED by impairing the endothelial integrity and function. At the same time, neutrophils are known to synthesize and store MMP-9 and MMP-25 [56], which are externalized during NETosis; thus, MMP-25 is supposed to produce a similar effect with MMP-9, enhancing endothelial MMP-2 activation and exacerbating ED and inflammation. Moreover, it has also been reported that both NE and MPO, as the components of NETs, can impair ECs [57,58]. NE can mediate the neutrophil-induced tissue damage and effectively degrade extracellular matrix components [2], while MPO is a heme protein in neutrophils and monocyte particles known to generate ROS [59]. On the other hand, NE and MPO can degrade tissue factor pathway inhibitors (TFPI), which restrains TF function as a major inhibitor of exogenous clotting pathways [51]. Therefore, damaging the anticoagulant function of TFPI can elevate TF expression, enhance coagulation and promote the occurrence of ED. On the contrary, the inhibition of NE by digesting NETs in different ways can markedly decrease NET-mediated cytotoxicity [2]. What is more, NE is believed to participate in the Endothelial-to-Mesenchymal Transition (EndoMT). EndoMT is a kind of special ED, NETs-associated NE that is alone believed to promote the nuclear translocation of junctional β-catenin and induce EndoMT in cultured ECs [60]. Moreover, the NETs-derived NE, together with MMP, attack the adhesion of ECs to adjacent structures by destroying the actin cytoskeleton, cadherin and VE-cadherin of ECs [50,58], which induce the transcription of β-catenin. Therefore, the activated β-catenin signaling further damages the vascular mucosal barrier and increases vascular endothelial permeability through aggravated EndoMT, and finally induce ED. Furthermore, experiment has shown that the MPO level was significantly raised in patients with coagulation disorders [61], and also the elevated MPO as an oxidase affects the endothelial function, phenotype and viability, ultimately leading to ED: (1) production through oxidants (HOCI, HOSCN, and NO_2_), (2) catalytic consumption of NO and (3) non-catalytic biological activity of recently discovered enzymes [59]. Cathepsin G, a serine protease abundant in NETs, was found to potentiate the effect of interleukin-1α (IL-1α) on the activation of ECs by cleaving the pro-IL-1α precursor and releasing the more potent mature IL-1α form [62], further acting on TF on the membrane. In general, a variety of specific enzymes in NETs can be significantly increased with elevated NETosis and impair the vascular functions through various pathways by aggravating ED and inflammatory response.

#### 2.4.3. NETs Induce the ED by Activating the Complement System

The complement system consists of approximately 30 serum-associated proteins and is classified as a part of the humoral innate immune system. Recently, it has been discovered that NETs can play a role in the initiation of complement system (Figure 4). The complement system is composed of a serine-protease cascade involving the continuous cleavage of complement proteins and forming MAC (C5b, C6, C7, C8, and C9) [63]. NETs can activate the complement system to amplify inflammation and disrupt the physiological function of ECs. Meanwhile, NETs-induced neutrophil-complement cascades may continue to attack surrounding ECs [51], and the complements located in inflammatory sites can further enhance the activation and recruitment of neutrophils and monocytes. Some complement effector factors synergistically interact with platelets to aggravate thrombotic inflammation, microvascular thrombosis and ED thrombotic microangiopathies [64,65,66]. For instance, C3a may activate platelets, while MAC and C5a may enhance TF expression in ECs, which reinforces the procoagulant activity and enhances the endothelial destruction [51]. Moreover, C5a can recruit and activate neutrophils with the upregulation of TLRs, complement receptors and other inflammatory receptors. Hua has demonstrated that the pre-stimulation of neutrophils with C5a can enhance the NETs release [67], and complement molecules in blood can be deposited on NETs to persistently function more [68]. In summary, the complement system activated by NETs can disrupt ECs, increase inflammation and further accelerate NETosis, hence promoting the vicious cycle of complement and NET-driven ED with thrombosis [69].

## 3. Diseases and Treatment

Although NETs play a defensive role in various diseases, they are active in inflammatory sites throughout the body. The disorder of NETosis can cause organic injury induced by ED, subsequently amplify the inflammatory response and accelerate the progression of disease, resulting in a vicious cycle between NETs, ED and inflammation with the uncontrolled situations. Therefore, it is essential to fully understand the intricated correlations between NETs and ED related with extensive diseases, including CVD, infectious and autoimmune disease; hence, the novel targets of therapeutics can be invented for the treatment of NETs-associated diseases. 

### 3.1. Cardiovascular Disease

CVD is a general term covering a series of ED-related diseases in the circulatory system, which are considered to be the main outcome of ED [70]. AS is the most typical one of CVD and has a great correlation with ED, which is regarded as a pathological basis of most CVD and a major cause of death worldwide [71,72]. Moreover, it is defined as a chronic inflammatory disease characterized by the lipid deposition with hyperplasia of vascular smooth muscle and fibrous matrix, and following AS plaques [73]. AS originates from ED with the accumulation and modification of low-density lipoproteins (LDLs) in the intima of blood vessels [74,75]. Subsequently, the modified LDLs, along with pro-inflammatory molecules, facilitate the activation of ECs and the recruitment of monocytes, resulting in local inflammation [72]. Thus, the activated ECs cause the disturbance of vascular endothelial environment and trigger inflammatory responses such as vasoconstriction, oxidative stress and lipid infiltration [34,76]. Obviously, the inflammation can induce ED. Conversely, ED can lead to inflammation. However, the NETs can promote AS by modulating both the ED and inflammation. Research found that NETs from SLE or Rheumatoid Arthritis (RA) patients were associated with ED and inflammation, which can accelerate AS [77,78]. Moreover, ED induces plaque instability, further leading to vascular occlusion, platelet aggregation and thrombosis [79]. AS plaques derived from ED divide into eroded and rupture-prone plaques. The eroded plaques are different from rupture-prone ones. They are rich in smooth muscle cells and proteoglycan such as hyaluronic acid and macrophages [50,80]. Therefore, triggers such as proteoglycan, erode plaques as DAMPs interact with TLR2 on the surface of ECs to activate ECs, which further promote the recruitment of neutrophils locally [51,81,82]. Subsequently, neutrophils at the inflammatory site degranulate and release ROS, resulting in the death and detachment of injured ECs [51]. Platelets and various clotting factors, such as von Willebrand factor (vWF) and P-selectin, can be activated in the mechanism. This leads to platelet aggregation and thrombosis [82,83]. On the other hand, the accumulation of neutrophil cholesterol, the exposure to LDL and oxidative stress around the plaques can initiate the recruited neutrophils to induce NETosis [84,85,86]. Moreover, P-selectin and HMGB-1 from activated platelets also interact with neutrophils to promote NETs release [87,88]. Furthermore, NETs further attack vascular endothelium by complement cascades and releasing enzymes such as MMP9 and NE and also aggravate vascular inflammation through DAMPs acting on TLR of ECs [51]. Moreover, the elevated NETs can interact with platelets to promote thrombosis [89]. In conclusion, ED induced NETs in AS plaques can act reversely on injury endothelium; therefore, NETs and ED together aggravate inflammatory damage and accelerate the process of AS.

Hitherto, AS is still one of the major diseases around the world with many serious complications. The familiar complication of AS is plaque rupture, the most common cause of death [90,91]. The ruptured plaques release their contents, including clot-forming substances such as tissue factors from macrophages and smooth muscle cells, exposing them to the bloodstream and triggering clots [92]. At the same time, the active neutrophils and NETs around the plaques further promote thrombogenesis and amplify endothelial injury [93]. Another serious complication is plaque erosion, characterized by increased smooth muscle cells and proteoglycan, which prevents plaque rupture and causes local vascular endothelial erosion [94,95,96]. Moreover, the surrounding NETs through TLR2 can aggravate the injury and detachment of ECs and aggravate, and hence strengthen, the local inflammatory response and thrombosis [50,97]. Therefore, the long-term and occluded thrombus may enhance the ECs’ damage in the vascular lumen, resulting in the local ischemic injury of vessels and tissues such as stroke and acute coronary syndrome [92]. In addition, this pathological process can deteriorate the development of peripheral vascular diseases such as severe ischemia of the lower extremities caused by thrombosis [92]. However, since the clinical prognosis of most patients has not reached expectations, in order to alleviate the pain and complications of patients, the therapy of AS should be further explored. For instance, the specific targeting of TLR2 or selective inhibition of complement C3 and C5 can inhibit the formation of NETs-induced local inflammation and thrombosis, further slow the progress of disease and the generation of thrombotic complications and eventually improve the prognosis of the disease.

The blockage of inflammation induced by NETosis recently became a novel therapeutic target of AS by alleviating ED. The inflammatory pathway is usually blocked by CI-amidine specifically with PAD4 inhibition, which is crucial for histone citrullination in NETosis; therefore, the raised ED will be prevented [98]. Moreover, DNase treatment is also efficient to the inhibition of ED-induced thrombosis to digest the DNA constitution of NETs (Table 1) [99]. In addition, roflumilast, the phosphodiesterase 4 inhibitor, can eliminate the interaction between NETosis and activated ECs and platelets [100]. Therefore, it looks possible to explore the importance of ED and NETs to arterial and venous thrombosis. This emphasizes the importance of NETs’ blockage to the prevention and treatment of ED-related disease.

### 3.2. Autoimmune Diseases

NETs are also verified to develop in autoimmune disease such as anti-neutrophil cytoplasmic antibodies (ANCA)-associated vasculitis (AAV) and systemic lupus erythematosus (SLE). It is supposed that NETs devote to disrupting self-tolerance of autoimmune diseases, which is regarded as the imbalance between the NET formation and NET degradation. Therefore, the long-term exposure of this abnormal NETs will continuously attack ECs and result in ED, which contribute to the vascular endothelial damage in various organs as complications, further impact the autoimmunity and accelerate the systemic organ damage during diseases.

#### 3.2.1. ANCA-Associated Vasculitis

ANCA-associated vasculitis (AAV) is a systemic autoimmune disorder given by the inflammatory reactions and small vessel destructions in multiple organs, which can appear at any age and affect 20–25 people in Europe each year [133,134]. AAV is characterized by continuous ED and inflammation of various blood vessels, with the rupture and occlusion of the vessels resulting in terminal organ damage and systemic inflammation. The disease is classified according to ANCA: p-ANCA targeting MPO and c-ANCA targeting proteinase 3 (PR3). Both antigens are derived from neutrophils with MPO in cytoplasm and PR3 on the cell membrane, which are released by NETosis [135,136]. Moreover, the turbulence of NETs regulation in the body impacts the vascular endothelium and vascular inflammation. Hence, the pathological basis of AAV is persistent aggravation of ED induced by NETs. When the body is infected by pathogens, the activated immune system produces pro-inflammatory factors to activate neutrophils, while the pathogens themselves also activate the defensive function of neutrophils, both ultimately leading to NETosis. However, patients with AAV have an inadequate ability to degrade NETs by DNase I [136,137], resulting in a large number of NETs circulating in the body for a long time. Meanwhile, MPO and PR3 antigens expressed by NETs can stimulate T cells to produce ANCAs and bind to their antigens, respectively [136]. The bindings further overactivate neutrophils, leading to the release of abnormal cytokines with ROS and more NETosis [138,139]. Thereafter, the elevated NETs cause constant damage to ECs through DAMPs and their enzymes, such as MMPs [112,118,137]. As a result, the persistent ED exacerbates systemic vascular inflammation and further vascular rupture. Subsequently, the ruptured endothelium stimulates platelet aggregation and thrombosis, which may lead to insufficient blood supply to terminal vessels and local necrosis within the organ [140]. In brief, AAV induces ED through the dysregulation of NETs, and constant ECs’ damage further leads to severe vascular inflammation in various organs.

Vascular inflammation of AAV commonly damage the kidneys and lungs [141,142]. Studies have shown that most kidney disease suffered by AAV patients may be life-threatening if not diagnosed [143]. Both human and animal studies have shown that neutrophils involved in the necrotizing inflammation resulting from extra- and vascular lesions of AAV; therefore, the abundant neutrophils are found in the glomerulus during early inflammation, and are subsequently replaced by macrophages [144]. Moreover, NETs also attack the ECs within the kidney, leading to further inflammation of the microvessels. On the other hand, the phenotype of AAV injuring lungs is divided into various types [142]. For instance, pulmonary capillaritis manifesting as diffuse alveolar hemorrhage (DAH) is a severe feature caused by the persistent destruction of alveolar capillary walls through the neutrophils disposition, breaking the integrity of the alveolar capillary membrane [145,146]. Eventually, the red blood cells infiltrate into the alveolar cavities, impairing the gas exchange process [146]. Although AAV-involved lesions of kidney and lung are common and death-related, the study of these complications has not been in-depth and few specific therapies have been explored for these lesions. Therefore, attention should be paid to early detection of such lesions in order to reduce organ damage and mortality and to finally improve prognosis.

Alleviation ED in AAV can be avoided through DNase I treatment to stop NETosis [112]. Moreover, the possible therapy of AAV is Intravenous Immunoglobulin (IVIG) (Table 1). It has been verified that the IVIG treatment of neutrophils before PMA exposure presents fewer NETs induced by PMA [123]; thus, the further ED could be evitable. In consequence, NETs could be served as a marker of AAV diagnosis and a target of AAV treatment by mitigate endothelial injury. However, in order to develop new drugs, the profound relationship between AAV and NETs should be further investigated.

#### 3.2.2. Systemic Lupus Erythematosus

SLE is classified as a typical autoimmune connective tissue disease, characterized by the production of a variety of autoantibodies that invade healthy tissues throughout the body [135]. Mild SLE is characterized by skin and joint lesions, while severe SLE is characterized by kidney and central nervous system damage, sometimes with catastrophic consequences. During the pathologic process of SLE, neutrophils are abnormally activated and produce a large number of ROS; thus, NETs can be found in SLE patients [147,148]. Various components of NET are regarded as antigens attacked by autoantibodies such as the externalized DNA and citrullinated histone H3 [3,149,150]. Moreover, SLE patients develop a unique type of neutrophils termed as low-density granulocytes (LDGs) circulating around the peripheral blood system [151,152]. These cells act as proinflammatory neutrophils to enhance the production of proinflammatory cytokines such as INF-II and have increased capacity to form NETs without any excited stimulation [3]. It is proved that NETs are the crucial source of autoantigens of SLE, and the deficiency of NETs’ degradation is served as a pathogenic factor [118]. What is more, the impaired clearance of NETs performs through the existence of DNase 1 inhibitors in NETs or the autoantibodies against protease after binding to NETs. Therefore, the raised NETosis and the production of autoantibodies can create a vicious cycle accelerating SLE progression. On the other hand, NETs may play a direct role in endothelial and tissue injury. MMP9 from NETs can activate MMP2 of ECs, resulting in ED and the apoptosis of ECs [56]. In addition, the excessive accumulation of NETs in blood vessels and glomerulus will enhance vascular permeability and EndoMT, which is associated with vascular disorder and organic tissue fibrosis [60]. In general, activated NETs in SLE can also cause ED, which in turn leads to systemic vascular inflammation, causing skin and mucosal bleeding and specific organ damage. More seriously, elevated ED in the kidney induces blood vessel occlusion, which leads to life-threatening kidney necrosis.

The skin and kidney lesions induced by SLE are often severe due to enhanced NETs deposition, possibly due to impaired NETs degradation mechanisms [3]. Skin is an important target organ of SLE autoimmune mechanism, and its mechanism may be multiple, with LL-37-related DNA NETs being the most typical [153,154]. It has been confirmed that a large amount of ds-DNA, LL-37 NETs, are found in the skin affected by human lupus [3]. Likewise, a similar pattern has been found in the kidneys of patients with lupus nephritis, where large deposits of NETs are associated with renal involvement and are accompanied by high levels of anti-ds-DNA antibodies [3,137]. Therefore, long-term exposure of excessive NETs as autoantigens in autoimmune conditions may lead to a continuous attack of autoantibodies and the formation of increased immune complexes, and further aggravate the injury and even the necrosis of vascular endothelium and organ tissues. Not limited to the skin and kidney, SLE is a disease that damages multiple organs throughout the body, and its poor prognosis will cause great pain to patients. Therefore, based on alleviating the pain of patients, the deeper research should be focused on the specific pathogenesis of NETs, such as selectively blocking the production of LDGs, in order to achieve the purpose of alleviating SLE progression and the treatment of complications.

SLE is not curable, so the appropriate treatment is closely related to the risk of death and patient outcome. On account of the above mechanisms, the SLE patients can be treated through modulating NETs. For instance, ED and CVD in SLE patients can be rescued by the inhibition of MMP [118]. Moreover, vitamin D (1, 25-(OH) 2-D3) stimulation of neutrophils could reduce the ED via decreasing the NETs’ activity (Table 1) [113,114]. In addition, metformin could repress the NETosis caused by PMA through regulating the mtDNA-pDC-IFNα pathway [122].

### 3.3. Sepsis

Sepsis refers to a systemic inflammatory response syndrome classified as a vital organ dysfunction caused by a dysregulated host response to infection, and it continues to be a major worldwide contributor to death [155]. In early sepsis, it undergoes a series of pathophysiological processes, mainly in the form of inflammation and coagulation dysfunction, with a high correlation with ED [156]. Moreover, there is a great correlation between NETs and sepsis. Recent reports indicate that the high levels of NETs can be detected when neutrophils are co-incubated with plasma of patients with sepsis [157]. In sepsis, the PAD-NET-CitH3 pathway activated by lipopolysaccharide (LPS) changes the function of pulmonary vascular endothelium [158]. In terms of inflammation resulting from sepsis, inflammation induces a large number of antibacterial components, such as cytokines, macrophages and neutrophils, to invade other inflammatory sites along the bloodstream, resulting in too many NETs around the bloodstream, which has a pathological effect on ECs to cause ED [62,140]. Additionally, circulating cell-free DNA (cfDNA), MPO-DNA and citrullinated histone H3 (Cit-H3) in NETs have referred to be associated with the level of organ dysfunction and 28-day mortality in septic shock patients [159,160,161]. At the same time, in sepsis, the excessive activation of NETs will cause microcirculatory disorders and ED, promote thrombosis and eventually lead to diffuse intravascular hemorrhage [162]. What is more, thrombosis and bleeding are related to vascular homeostasis, and endothelial state is a major factor in maintaining vascular homeostasis. Therefore, NETs-induced ED of sepsis can cause serious complications. 

In terms of complications, sepsis mainly induces septic shock, pulmonary dysfunction, kidney injury, heart injury, blood coagulation dysfunction and so on [156,163,164,165]. At present, sepsis-induced lung injury is closely related to NETs. Some experiments have pointed out that NETs are overexpressed in patients with ARDS/ALI [163,166]. In addition, recent studies have shown that exocrine secretion from platelets over-increases NETs’ production in septic shock [167]. If these complications progress to a severe irreversible stage, it will greatly increase the mortality rate of patients [168]. Because of the prevalence and high mortality rate of sepsis, scientists began to study the treatment and prognosis of sepsis decades ago. However, as systemic sepsis is often severe and irreversible, it is particularly important to capture early signs of sepsis and delay the process of sepsis. As mentioned earlier, NETs and relevant molecules are practical targets for the control of sepsis. However, the efficacy of drugs is often difficult to balance with symptoms. Therefore, continuous research to find a suitable drug can not only effectively inhibit NETs but also avoid aggravating other symptoms, and this can be considered a breakthrough.

Although there is no powerful drug to regulate NETs-induced ED to treat sepsis, many researchers have devoted themselves to finding NETs-related drugs to alleviate the development of sepsis. Statins and angiotensin receptor blockers (ARBs) have been found to block the activation of TLR4 and may be associated with relieving ECs destruction. Especially, fluvastatin, simvastatin and atorvastatin are effective inhibitors [169]. However, by the time sepsis is diagnosed, the inflammatory response in the body may be difficult to contain, and it is particularly important to identify early signs of sepsis as early as possible. In fact, early detection and the relief of ED can prevent many serious conditions in the later stages of the disease. Schattner suggests that the inhibition of TLR4 can improve the prognosis of sepsis and may be a novel way to treat sepsis since TLR inhibitors can prevent components from NETs binding to TLRs with ED [170]. However, it is noteworthy that early blocking of TLR4 may also reduce the immunity to pathogen toxins. Therefore, clinically, the advantages and disadvantages of this kind of drugs mean that we should be cautious [68]. Moreover, some researchers have mentioned that PAD4 is a key factor in the formation of NETs [171]. The inhibition of PAD4 to alleviate the ED may be another way to treat sepsis since PAD4 knockout mice had higher survival and prognosis in septic mice [172]. Similarly, PAD4 inhibitors BB-Cl-amidine can relieve NETs-related vascular endothelial injury to some extent [173]. Recently, a novel finding suggested that PAD4 inhibitor GSK484 can also inhibit NETs-induced thrombosis by affecting kindlin-3 in neutrophils [174]. Therefore, it is hypothesized that PAD4 inhibitors have a great relevance to the prevention of ED. Additionally, although there is still much controversy about heparin in the treatment of NETs-induced thrombosis in patients with sepsis, heparin is undoubtedly a powerful choice for anticoagulation. It has been pointed out that low concentration of heparin (250 U/kg) has an inhibitory effect on NETs, while the high concentration of heparin can over-activate the coagulation system and directly activate NETs [175]. Hence, when heparin is used to treat sepsis clinically, doctors need to determine the dose to avoid activating NETs to fasten the development of sepsis. Moreover, other drugs and their mechanisms are shown in the table (Table 1).

### 3.4. Others

In addition to the diseases mentioned above, the interaction between NETs and ED can make sense in cancer by promoting tumor metastasis. The primary tumor site can release extracellular vesicles (EVs), various ROS generating proinflammatory factors and specific pro-NETotic factors into the circulation [176]. The activated neutrophils produce ROS inducing local inflammation and further facilitate NETosis. Meanwhile, TF released by cancer cells activates platelets with HMGB1 release, and also further promotes NETs through producing ROS [87,177]. What is more, the tumor-derived EVs may stimulate neutrophils to phagocytose tumor membrane fragments and encapsulated factors in order to release NETs. Moreover, the proteolytic components of NETs, such as NE and MPO, contribute to ED, causing the release of inflammatory factors such as IL-8 and further promoting the recruitment of neutrophils and NETosis [176]. In addition, under the infiltration of cancer, cancer cells promote the overexpression of granulocyte colony-stimulating factor (G-CSF), which binds to corresponding receptors on the cell surface to activate neutrophils [178,179,180]. In a word, the tumor-released factors, activated platelets, and activated endothelium interact with the receptors on neutrophils, respectively, resulting in NETs release. Moreover, NETs can also trap the circulating cancer cells and facilitate the metastases of tumor lesions [176]. Therefore, the detailed NETs’ mechanisms will be effective in the control of cancer progression as specific targets.

Diabetes is a metabolic disorder that is prevalent in the world [181,182]. According to WHO, the number of people with diabetes is predicted to reach 366 million by 2030 [183]. Patients with diabetic ophthalmopathy and diabetic kidney disease have higher mortality rates than high-risk diseases such as tuberculosis and AIDS [184]. The biggest sign of diabetes is elevated blood sugar. A large number of studies have shown that abnormal glucose can imbalance the microvascular environment, destroy the blood-retinal barrier during the proliferation of diabetic retinopathy (DR) and lead to fluid exudation and retinal hemorrhage [185,186]. Previous studies have shown that markers of NETs are increased in the serum and eye tissues of diabetic patients [183,187], and hyperglycemia can induce the formation of internal and external NETs [183]. Recent studies have shown that NETs are involved in the pathological processes of many eye diseases [188,189]. In DR, hyperglycemia-induced NADPH can promote the production of ROS and further facilitate NETosis and vascular endothelial growth factor (VEGF) release, which lead to high concentrations of NETs’ deposition in the retina and vitreous, resulting in increased ocular inflammation and microcirculation disorders [183]. Therefore, the above process provides the hints of anti-DR therapy targets. It is reported that anti-VEGF drugs have been shown to treat DR by binding to vascular endothelial specific receptors (VEGFR) [190].

## 4. Conclusions

Both the triggers of NETs and ED inevitably interrelate to the inflammatory response while the body is injured. Increased studies demonstrated, through the detailed mechanisms of their correlations, that NETs and ED can interact together with inflammation as a mediator, and further amplify the local inflammation, even making it systemic. The intricate pathway serves as a body defense mechanism. However, it will damage surrounding tissue and organs if dysregulated. Therefore, the specific molecules within the above pathways are promising as therapeutic targets to alleviate and block the disordered inflammations for the treatment of relevant disease.

## Figures and Tables

**Figure 1 ijms-23-05626-f001:**
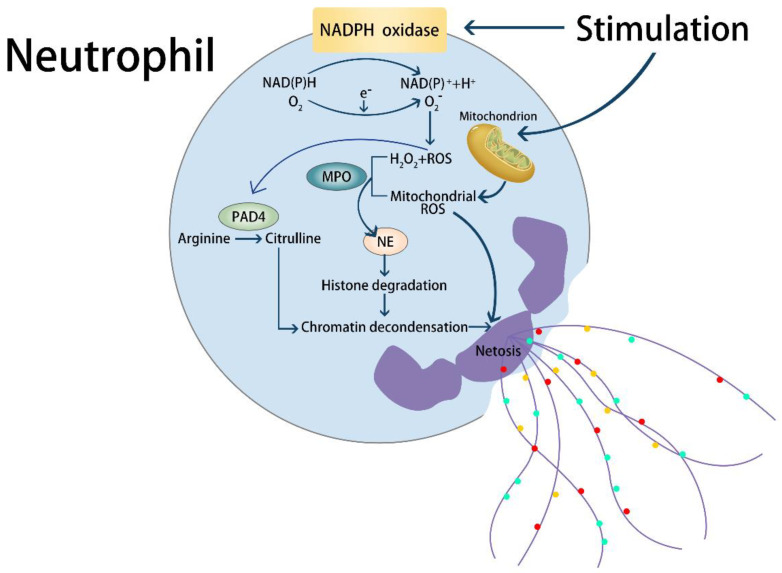
The biomolecular mechanism of NETosis. Extracellular stimulation activates corresponding on-membrane receptors, thereby releasing NETs through different intracellular pathways. During NETosis, the activated neutrophils secrete reactive oxygen species (ROS) via nicotinamide adenine dinucleotide phosphate (NADPH) or mitochondria, which then activates myeloperoxidase (MPO), further facilitating the release of neutrophil elastase (NE) from cytoplasmic particles and the translocation into the nucleus. Nucleic NE degrades histones and disrupts the assembly of chromatin in the nucleus, resulting in chromatin. At the same time, peptidyl arginine deiminase 4 (PAD4), locates the downstream of ROS, catalyzes the deamination and citrullination of histone arginine residues and then promotes chromatin decondensation, which is the crucial point within NETosis.

**Figure 2 ijms-23-05626-f002:**
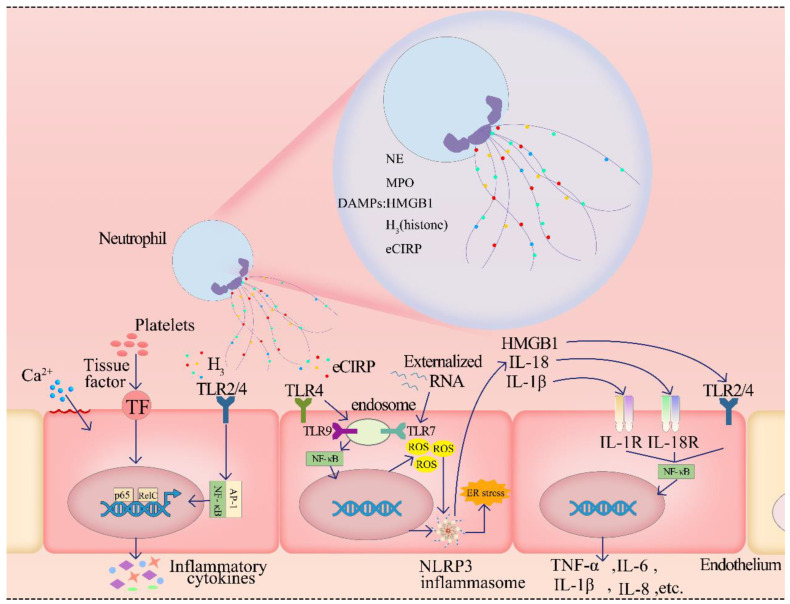
After neutrophil formation of NETs, damage-associated molecular patterns (DAMPs) are released by NETs. Histones selectively bind to Toll-like receptors (TLR) at different concentrations, leading to the activation of NF-κB and the transcription of AP-1. On the one hand, this may cause ECs to release inflammatory cytokines to amplify the inflammatory response. On the other hand, it promotes the expression of tissue factor (TF) and leads to platelet aggregation. Finally, all of them cause damage to the function of ECs. In addition, the binding of histone to EC membrane can also cause cell membrane perforation, lead to calcium ion inflow into cells and damage ECs. Extracellular histone, extracellular cold-inducible RNA-binding protein (eCIRP), as another molecule of DAMPs, activate TLR4/TLR9/NF-κB signal pathway through NETs release and binding to EC surface receptors. After ROS and NLRP3 inflammasome are released to activate the pathway, NLRP3 can be activated by ROS to cause ER stress and secrete pro-inflammatory substances (IL-18, IL-1β, HMGB1). Pro-inflammatory substances promote the release of more pro-inflammatory granules (TGF-α, IL-6, IL-1β, IL-8) and further aggravate ED by activating NF-κB signal pathway by binding to the corresponding surface receptors of peripheral ECs. IL-8 secreted by ECs can also act on more neutrophils, increasing the expression of NETs, forming positive feedback and aggravating NETosis. In another novel way, NETosis can release externalized RNA. Externalized RNA is internalized by the surrounding ECs by binding to TLR9, resulting in ED.

**Figure 3 ijms-23-05626-f003:**
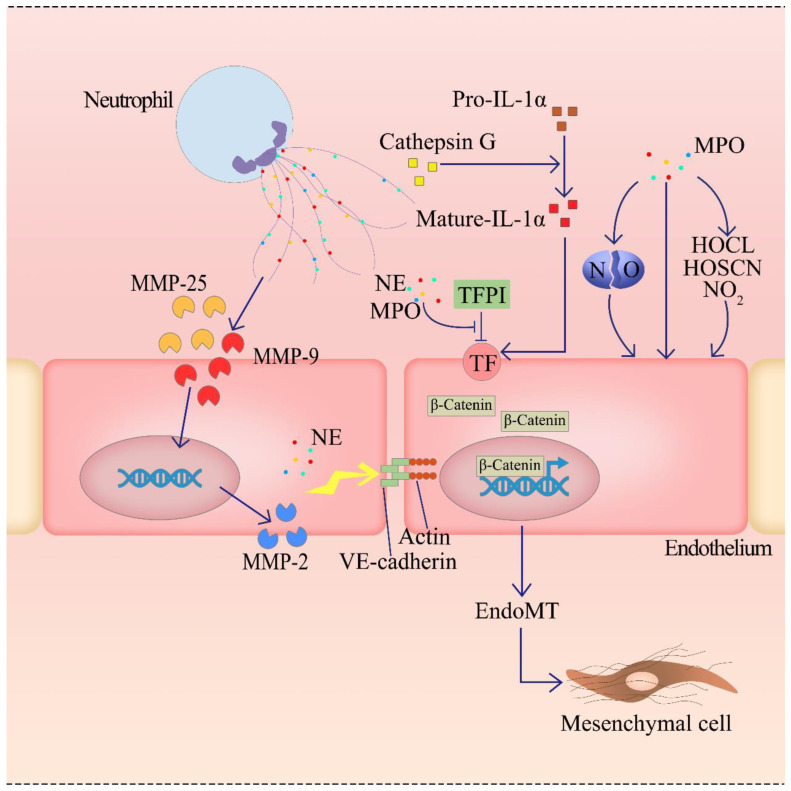
The metalloproteinase-9 (MMP-9) (may together with MMP-25) released after the production of NETs will interact with ECs, causing ECs to produce MMP-2. MMP-2 and NE from NETs to attack the junction structures between ECs—VE-cadherin, E-cadherin and actin, resulting in damage to the physiological structure of ECs. Meanwhile, net-induced β-catenin nuclear translocation induces Endothelial-to-Mesenchymal Transition (EndoMT) in ECs, which can be exacerbated by impaired VE-cadherin. In addition, neutrophil elastase (NE) and myeloperoxidase (MPO) were released by NETosis to destroy tissue factor pathway inhibitor (TFPI). By inhibiting the decomposition of TF, the expression level of TF was increased, and the endothelium was damaged by increasing blood viscosity. MPO derived from NETs directly destruct ECs by breaking NO, oxidants (HOCI, HOSCN, and NO_2_) production and non-catalytic pathway. Moreover, Cathepsin G cleaves the pro-IL-1α precursor into interleukin-1α (IL-1α) and further acts on TF.

**Figure 4 ijms-23-05626-f004:**
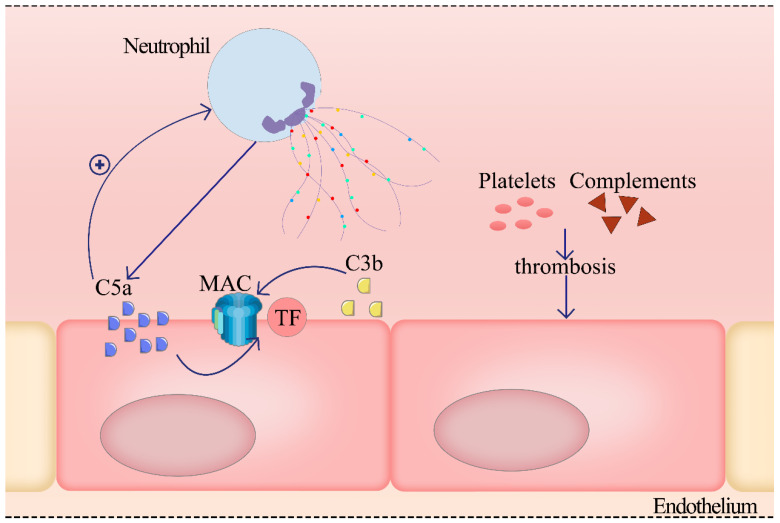
NETs-derived C3b activates the complement cascade, which amplifies the inflammatory response by activating neutrophils. The interaction of membrane attack complex (MAC) with C5a enhances TF expression in ECs and destroys ECs. Some factors in the complement system interact with platelets to enhance procoagulability. More neutrophils will be recruited by complement stimulation, and complement can also be deposited on NETs to continue to function.

**Table 1 ijms-23-05626-t001:** Summary of therapeutic strategy targeting neutrophils.

Disease	Drugs	Strategies	Outcome	Reference
Cardiovascular Disease	DNase 1	DNA degradation	Digest the DNA constitution of NETs, therefore destruct the NETs, protected murine IVC stenosis model from DVT	[99,101]
	CI-amidine	PAD4 inhibitor	Block the histone citrullination in NETosis to reduce NETosis and eliminate inflammation in DIO mice	[98,101]
	Roflumilast	Phosphodiesterase 4 inhibitor	Eliminate the interaction between NETosis and activated ECs and platelets in order to prevent platelet aggregation	[100,102]
	Heparin	Anti-histone	Block the histone-induced NF-κB pathway, thus protect the ECs from inflammation of NETs, therefore avoid mice from organ damage	[103,104,105,106]
	Anti-high-mobility group box 1 (HMGB1) pAb	Anti-HMGB1	Reduce the histone 3 and free DNA in the BAL fluid of LPS-treated mice, thus decrease the inflammation and neutrophil chemotaxis to mitigate NETosis	[107,108,109,110,111]
Autoimmune Diseases	DNase	DNA degradation	Digest the DNA constitution of NETs, therefore avoid glomerular endothelial injury in murine AAV disease models	[101,112]
	Vitamin D	Inhibiting NETs activity	Decrease the NETs activity to reduce the damage to ECs, and reduce the early cellular apoptosis in SLE patients	[113,114]
	Chloroquine/ Hydroxychloroquine (HDQ)	MMPs-TIMPs modulation	Modulate NETs through the regulation of MMP and TIMP to maintain the extracellular homeostasis in SLE patients; also it can prevent platelet aggregation, resulting in endothelium protection	[115,116,117,118,119]
	Metformin	Regulating mtDNA-pDC-IFNα pathway	Inhibit ROS production, and repress NETosis with a reduction in elastase, proteinase-3, histones, and cfDNA with in chronic autoimmune disease of the elderly	[120,121,122]
	Intravenous Immunoglobulin (IVIG)	Inhibiting ANCA production	Relieve antigen antibody responses, and inflammation, therefore NET amounts in the peritoneum are significantly decreased	[123,124]
Sepsis	Drotrecogin	Recombinant human activated protein C	Inhibit the formation of coagulation factors Va and VIlla and destroy extracellular histones, preventing activated platelets from inducing NETosis	[125,126,127]
	LL-37	Enhancing NETs	Improve sterilization capacity and increase the survival rate of CLP mice	[128]
	DNase I	DNA degradation	Combine with antibiotics to improve the outcome	[129]
	Anti–TREM-1	Reducing NETosis	Eliminate associated ED and organic damage in mice LPS models	[130]
	Small Polyanions (SPAs)	Histone inhibitor (NET-bound and free)	Improve the outcome in the LPS, TNF and CLP mice models	[131]
	Defibrotide	Neutralization of histones (cationic proteins) with polyanionic compounds	In vitro, defibrotide counteracted EC activation and pyroptosis-mediated cell death induced by NETs. In vivo, defibrotide stabilized the endothelium and protected against histone-accelerated inferior vena cava thrombosis in mice. The development of MODS was relieved in the later stage of sepsis	[132]

## Data Availability

The data presented in this study are available in article.

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
