# Peer review of "Endothelial Dysfunction Induced by Extracellular Neutrophil Traps Plays Important Role in the Occurrence and Treatment of Extracellular Neutrophil Traps-Related Disease"

_ijms, 2022, doi:10.3390/ijms23105626_

Round 1

Reviewer 1 Report

The Current review manuscript submitted by Shuyang Yu et al primarily focused on activation of inflammatory neutrophils and release of extracellular traps leads to endothelial dysfunction. Authors mentioned several mechanisms and pathways involved in the damage of Endothelial cells by NETS.  In the circulation, neutrophils are confirmed to have a significant impact on endothelial function by damaging endothelial cells (ECs) directly and their surrounding structures. Furthermore, neutrophils also can release a particular structure termed extracellular neutrophil traps (NETs), which can induce the local inflammation of ECs and further cause endothelial dysfunction (ED). ED are involved in many diseases, including CVD, Autoimmune Diseases, SLE, Sepsis, COVID-19 etc. The current review manuscript is written very well and broadly discussed the Linkage between NETS and ED together with mechanism behind the NET formation and associated diseases along with treatment strategies. However, authors need to address several comments and suggestions before the manuscript get accepted in IJMS, peer reviewed journal of MDPI.

Comments:

  1. It would be very additive and novel to the current review if the authors could discuss the role of NETs inducing the ED under the Diabetic setting and Diabetic kidney disease, Chronic Kidney disease, bone mineral disorder and inflammatory bowel disease.
  2. What are the future challenges and clinical complications? It would better f authors could write the section and discuss about the potential challenges together with its associated clinical complications.  
  3. Authors also addressed the role of NETs and endothelial dysfunction. This section is little ambitious as there is not much strong evidences/studies supporting the direct relationship between NETs associated ED and Covid-19 complications. I would suggest to delete the Covid-19 section which allow the readers focused much more evidence based, not a hypothetical.

Author Response

Response to the reviewers

The Current review manuscript submitted by Shuyang Yu et al primarily focused on activation of inflammatory neutrophils and release of extracellular traps leads to endothelial dysfunction. Authors mentioned several mechanisms and pathways involved in the damage of Endothelial cells by NETS.  In the circulation, neutrophils are confirmed to have a significant impact on endothelial function by damaging endothelial cells (ECs) directly and their surrounding structures. Furthermore, neutrophils also can release a particular structure termed extracellular neutrophil traps (NETs), which can induce the local inflammation of ECs and further cause endothelial dysfunction (ED). ED are involved in many diseases, including CVD, Autoimmune Diseases, SLE, Sepsis, COVID-19 etc. The current review manuscript is written very well and broadly discussed the Linkage between NETS and ED together with mechanism behind the NET formation and associated diseases along with treatment strategies. However, authors need to address several comments and suggestions before the manuscript get accepted in IJMS, peer reviewed journal of MDPI.

Comments:

1.It would be very additive and novel to the current review if the authors could discuss the role of NETs inducing the ED under the Diabetic setting and Diabetic kidney disease, Chronic Kidney disease, bone mineral disorder and inflammatory bowel disease.

Reply: Dear reviewer, thank you very much for your approval and suggestions. After our literature search on Pubmed, there are almost no studies on the relationship between neutrophil-Nets-ED in diabetic nephropathy, chronic kidney disease, bone mineral disorders and inflammatory bowel disease. We are sorry for not being able to summarize these aspects in our manuscript. However, after our search, we found a large number of diabetic retinopathy content consistent with the manuscript theme, so we added paragraphs.(Lines553-570), the detailed contents as follow:

Diabetes is a metabolic disorder that is prevalent in the world [181,182]. Accord-ing to WHO, the number of people with diabetes is predicted to reach 366 million by 2030 [183]. Patients with diabetic ophthalmopathy and diabetic kidney disease have higher mortality rates than high-risk diseases such as tuberculosis and AIDS [184]. The biggest sign of diabetes is the elevated blood sugar. A large number of studies have shown that abnormal glucose can imbalance the microvascular environment, destroy the blood-retinal barrier during the proliferation of diabetic retinopathy (DR), and lead to fluid exudation and retinal hemorrhage [185,186]. Previous studies have shown that the markers of NETs are increased in the serum and eye tissues of diabetic patients [183,187], and hyperglycemia can induce the formation of internal and external NETs [183]. Recent studies have shown that NETs are involved in the pathological processes of many eye diseases [188,189]. In DR, hyperglycemia-induced NADPH can promote the production of ROS, and further facilitate NETosis and vascular endothelial growth factor (VEGF) release, which lead to the high concentrations of NETs deposition in the retina and vitreous, resulting in the increased ocular inflammation and microcircula-tion disorders [183]. Therefore, the above process provides the hints of anti-DR therapy targets. It is reported that anti-VEGF drugs have been shown to treat DR by binding to vascular endothelial specific receptors (VEGFR) [190].

2.What are the future challenges and clinical complications? It would better f authors could write the section and discuss about the potential challenges together with its associated clinical complications. 

Reply: Dear reviewer, thank you very much for your approval and suggestions, and we have added paragraphs in section 3 to discuss the complications and potential challenges of the diseases in our manuscript(Lines342-362;399-415; 449-464;493-506)

3.Authors also addressed the role of NETs and endothelial dysfunction. This section is little ambitious as there is not much strong evidences/studies supporting the direct relationship between NETs associated ED and Covid-19 complications. I would suggest to delete the Covid-19 section which allow the readers focused much more evidence based, not a hypothetical.

Reply: Dear reviewer, thank you very much for your approval and suggestions. Initially, when we searched the literature, we found that there was a large literature stating the relationship between NETs and COVID-19, so we added the COVID-19 section. However, after thinking about it, we found that because this is an emerging disease, both in vitro and in vivo experiments and drug experiments are insufficient. So we deleted that part.

Reviewer 2 Report

The authors have submitted a review article of illustrating a current knowledge regarding impact of extracellular neutrophil traps (NETs) on endothelial dysfunction casing inflammation-related diseases such as atherosclerosis, ANCA-associated vasculitis, sepsis, and symptoms of COVID-19 (demonstrated in Table 1). The authors searched a range of eligible literature, from well-known classical, and latest research regarding an association of cellular components which are responsible for the NETS with the related endothelial dysfunction, which are primarily attributed to the endothelial diseases described in this review. The authors discussed the beneficial availability of the regulation of neutrophil function and the pharmacologic properties which ameliorate the states of the disease situation, resulting in reliable perspectives. This issue is of interest, and impact of their review is strong. My overall concern with the review describing the current available data regarding beneficial availability of the compounds listed in this review against various endothelial diseases is that information provided may offer something substantial that helps advance our understanding of effective management which draws novel class of effective compounds available in clinic. The reference list may be useful for readers who are interested in this issue.

To strengthen authors’ perspectives, the authors are strongly recommended to add a “physiologic relevance of NETs which are primarily responsible for both good aspects of foreign infection and also bad aspects of endothelial dysfunction in host body”. A rational understanding of these conflicting facts, if known, may support largely the authors’ perspective.

Author Response

Response to the reviewers

The authors have submitted a review article of illustrating a current knowledge regarding impact of extracellular neutrophil traps (NETs) on endothelial dysfunction casing inflammation-related diseases such as atherosclerosis, ANCA-associated vasculitis, sepsis, and symptoms of COVID-19 (demonstrated in Table 1). The authors searched a range of eligible literature, from well-known classical, and latest research regarding an association of cellular components which are responsible for the NETS with the related endothelial dysfunction, which are primarily attributed to the endothelial diseases described in this review. The authors discussed the beneficial availability of the regulation of neutrophil function and the pharmacologic properties which ameliorate the states of the disease situation, resulting in reliable perspectives. This issue is of interest, and impact of their review is strong. My overall concern with the review describing the current available data regarding beneficial availability of the compounds listed in this review against various endothelial diseases is that information provided may offer something substantial that helps advance our understanding of effective management which draws novel class of effective compounds available in clinic. The reference list may be useful for readers who are interested in this issue.

To strengthen authors’ perspectives, the authors are strongly recommended to add a “physiologic relevance of NETs which are primarily responsible for both good aspects of foreign infection and also bad aspects of endothelial dysfunction in host body”. A rational understanding of these conflicting facts, if known, may support largely the authors’ perspective.

Reply: Dear reviewer, thank you for your review, we have added some physiologic relevance of NETs in section 2.3 to discuss good and bad aspects (Lines 118-126), the detailed contents as follow:

Neutrophils produced NETs can protect the host from pathogens including most gram-negative bacteria, gram-positive bacteria, viruses, parasites and so on [25]. First of all, it can prevent the spread of infections through the extracellular DNA network from NETs [10]. Second, it will kill pathogens through its own ingredients, for instance, NE plays a role through slicing the virulence factors [10]. However, in addition to host protection function, when it exists for a long time or accumulates in large quantities, it will initiate a variety of pathological processes related to endothelial damage, and eventually cause permanent damage to ECs [26]. Studies have shown that the histones, MPO,NE and Cathepsin G are the primary components of NETs engaged tissue destruction [27].

Reviewer 3 Report

The author's provided a very comprehensive review manuscript related to the function of NET formation promoting EC dysfunction and their effects on several inflammatory diseases (like cardiovascular diseases, AAV, SLE, sepsis). The manuscript is well written; however, a newly topic related to neutrophil adaptive (memory-like) reactions and their putative effects have been not discussed. Since it is of major interest for the scientific community I would suggest to include a new paragraph in regard to this topic.

Here some interesting published works from different groups that are working on this field:

  1. Lajqi, T.; Braun, M.; Kranig, S.A.; Frommhold, D.; Pöschl, J.; Hudalla, H. LPS Induces Opposing Memory-like Inflammatory Responses in Mouse Bone Marrow Neutrophils. Int. J. Mol. Sci. 202122, 9803. (this paper shows the role of pathogen dose promoting adaptive manners in neutrophils); 
  2. Moorlag, S.J.C.F.M.; Rodriguez-Rosales, Y.A.; Gillard, J.; Fanucchi, S.; Theunissen, K.; Novakovic, B.; de Bont, C.M.; Negishi, Y.; Fok, E.T.; Kalafati, L.; et al. BCG vaccination induces long-term functional reprogramming of human neutrophils. Cell Rep. 202033, 108387. (this paper shows that trained neutrophils express increased antimicrobial activity)
  3. Kalafati, L.; Kourtzelis, I.; Schulte-Schrepping, J.; Li, X.; Hatzioannou, A.; Grinenko, T.; Hagag, E.; Sinha, A.; Has, C.; Dietz, S.; et al. Innate immune training of granulopoiesis promotes anti-tumor activity. Cell 2020183, 771–785. (this paper shows that trained neutrophils express a ROS-dependent anti-tumoral activity)
  4. Gut Microbiota-Derived Small Extracellular Vesicles Endorse Memory-like Inflammatory Responses in Murine Neutrophils. Biomedicines 202210, 442. (this paper shows that bacterial EVs may promote also the induction of trained immunity as adaptive manner of neutrophils)
  5. and a newly preprint which is also quite interesting in reagrd to the topic "Gram-posivite Staphyloc. aureus LTA promotes dinstinct memory-like effects in murine bone marrow neutrophils" (which interesting shows that even gram-positive bacteria may support different memory-like inflamamtory responses in neutrophils).

There are also increasing number of studies in the filed and you don't have to stick to my recommendations mentioned above, but I find them interesting and would further increase the reader interest for your review.

Author Response

Response to the reviewers

The author's provided a very comprehensive review manuscript related to the function of NET formation promoting EC dysfunction and their effects on several inflammatory diseases (like cardiovascular diseases, AAV, SLE, sepsis). The manuscript is well written; however, a newly topic related to neutrophil adaptive (memory-like) reactions and their putative effects have been not discussed. Since it is of major interest for the scientific community, I would suggest to include a new paragraph in regard to this topic.

Here some interesting published works from different groups that are working on this field:

  1. Lajqi, T.; Braun, M.; Kranig, S.A.; Frommhold, D.; Pöschl, J.; Hudalla, H. LPS Induces Opposing Memory-like Inflammatory Responses in Mouse Bone Marrow Neutrophils. Int. J. Mol. Sci. 202122, 9803. (this paper shows the role of pathogen dose promoting adaptive manners in neutrophils); 
  2. Moorlag, S.J.C.F.M.; Rodriguez-Rosales, Y.A.; Gillard, J.; Fanucchi, S.; Theunissen, K.; Novakovic, B.; de Bont, C.M.; Negishi, Y.; Fok, E.T.; Kalafati, L.; et al. BCG vaccination induces long-term functional reprogramming of human neutrophils. Cell Rep. 2020, 33, 108387. (this paper shows that trained neutrophils express increased antimicrobial activity)
  3. Kalafati, L.; Kourtzelis, I.; Schulte-Schrepping, J.; Li, X.; Hatzioannou, A.; Grinenko, T.; Hagag, E.; Sinha, A.; Has, C.; Dietz, S.; et al. Innate immune training of granulopoiesis promotes anti-tumor activity. Cell 2020, 183, 771–785. (this paper shows that trained neutrophils express a ROS-dependent anti-tumoral activity)
  4. Gut Microbiota-Derived Small Extracellular Vesicles Endorse Memory-like Inflammatory Responses in Murine Neutrophils. Biomedicines 202210, 442. (this paper shows that bacterial EVs may promote also the induction of trained immunity as adaptive manner of neutrophils)
  5. and a newly preprint which is also quite interesting in reagrd to the topic "Gram-posivite Staphyloc. aureus LTA promotes dinstinct memory-like effects in murine bone marrow neutrophils" (which interesting shows that even gram-positive bacteria may support different memory-like inflamamtory responses in neutrophils).

There are also increasing number of studies in the filed and you don't have to stick to my recommendations mentioned above, but I find them interesting and would further increase the reader interest for your review.

Reply: Dear reviewer, thank you for your review, we have added these contents about memory-like response of neutrophils in section 2.1 in our manuscript (Lines 49-62), the detailed contents as follow:

Interestingly, recent studies have suggested that neutrophils may also have a memory-like immune mechanism against infection, termed neutrophil adaptive (memory-like) reactions [6-9]. For instance, Bacillus Calmette-Guerrin (BCG) vaccine and β-glucan can induce adaptive response of naive neutrophils [7,8], which leads to reprogramming of neutrophil transcriptome and epigenetics modifications in tri-methylation at histone 3 lysine 4, and finally the release of pro-inflammatory mediators. As a result, when neutrophils are similarly stimulated for the second time, they can be heavily recruited and activated to mediate a more intense immune response. Mean-while, an interesting study found that the process may be concentration-dependent [6,9]. Low dose of microbiota-derived components (LPS or Small Extracellular Vesicles) have been shown to induce adaptive response of neutrophils, while high dose of microbiota-derived components can inhibit it [9]. Mechanistically, alterations in TLR2/MyD88 as well as TLR4/MyD88 signaling were correlated with the induction of adaptive cues in neutrophils in vitro.

Round 2

Reviewer 1 Report

The current version of the manuscript is revised and resubmitted review manuscript. This is very excellent review, describing the relationship between the Ed and Netosis, explored the novel mechanism which can be used as targeted therapy for translations standpoint. Authors edited, addressed and provided sufficient information in response to all the comments raised by reviewers. The current version can be accepted as Publication for IJMS, peer reviewed journal of MDPI.  Congratulation to authors.  

Thank you very much to Editor of IJMS for involving me review this study.

Reviewer 2 Report

The authors have addressed properly all the issues raised by reviewers including me. I have no more comments, and now recommend that this manuscript is acceptable for publication in the IJMS journal.